# Preparation and Study of Photocatalytic Properties of (M(M=Pt, Ag and Au)-TiO$_2$)@MoS$_2$ Nanocomposites

**Liying Ju** [1,†], **Dunhua Hong** [1,*,†], **Xing Jin** [1], **Hongxian Liu** [1], **Xiude Yang** [1] , **Liying Nie** [1], **Qibin Liu** [2], **Zhixi Gao** [1], **Wei Zhu** [1], **Yi Wang** [1] and **Xiang Yang** [1]

1  School of Physical and Electronic Science, Zunyi Normal College, Zunyi 563006, China
2  School of Materials and Metallurgy, Guizhou University, Guiyang 550025, China
*  Correspondence: dunhua_2008@yeah.net
†  These authors contributed equally to this work.

**Abstract:** There have been many articles on the degradation of pollutants by binary and ternary nanocomposites in the field of photocatalysis. However, there has been no research comparing the photocatalytic performance of Rhodamine B (Rh B) between (M(M=Pt, Ag and Au)-TiO$_2$)@MoS$_2$ nanocomposites and binary nanocomposites. To this end, we prepared and studied (M(M=Pt, Ag and Au)-TiO$_2$)@MoS$_2$ nanocomposites and compared their photocatalytic degradation efficiency with binary composites and parent materials for Rhodamine B. We concluded that the best ternary polymer nanocomposite for degrading Rhodamine B is (Pt(5 wt%)-TiO$_2$(15 wt%))@MoS$_2$. In this work, a series of MoS$_2$, TiO$_2$@MoS$_2$, and (M(M=Pt, Ag and Au)-TiO$_2$)@MoS$_2$ nanocomposites with various compositions were synthesized by the hydrothermal and deposition–precipitation methods, and their photocatalytic characteristics were studied in depth using X-ray diffraction (XRD), scanning electron microscopy (SEM), transmission electron microscopy (TEM), X-ray photoelectron spectroscopy (XPS) photoluminescence spectra (PL), FTIR spectra, UV–Vis DRS spectra, and BET analyzer. The results confirmed that TiO$_2$ and M(Pt, Ag and Au) nanoparticles (NPs) were evenly distributed on MoS$_2$ nanosheets (NSs) to form (M(M=Pt, Ag and Au)-TiO$_2$)@MoS$_2$ nanocomposite heterojunction. The UV–Vis absorption spectrum test results indicated that (Pt(5 wt%)-TiO$_2$(15 wt%))@MoS$_2$ ternary heterojunction nanocomposites exhibited the highest photocatalysis activity, with the maximum value of 99.0% compared to 93% for TiO$_2$(15 wt%)@MoS$_2$, 96.5% for (Ag(5 wt%)-TiO$_2$(15 wt%))@MoS$_2$, and 97.8% for (Au(5 wt%)-TiO$_2$(15 wt%))@MoS$_2$ within 9 min. The advanced structure of (Pt-TiO$_2$)@MoS$_2$ improved both light harvesting and electron transfer in the photocatalytic composites, contributing to remarkable catalytic effectiveness and extended durability for the photodegradation of Rhodamine B (Rh B). In-depth discussions of the potential growth and photocatalytic mechanism, which will help improve the energy and environmental fields, are included.

**Keywords:** photocatalysis; MoS$_2$; ternary heterojunction nanocomposites; degradation

## 1. Introduction

Currently, the concern regarding natural contamination has increased overall research interests in the photocatalysis process, which is by and large viewed as one of the best high-level oxidation processes [1,2]. Organic contaminants can be degraded using photocatalytic degradation technology into non-toxic carbon dioxide and water directly via solar energy, which is a non-polluting green wastewater treatment technology [1–3]. MoS$_2$, as a two-dimensional nanomaterial, has a band gap of its nanostructure of around 1.78 eV and has photocatalysis activity under ultraviolet and visible light, so is considered to be a promising candidate [4]. Therefore, a type of MoS$_2$ nanocomposite catalytic material could be developed and achieve excellent photocatalysis results.

In the last few decades, studies have shown that MoS$_2$ composites loaded with precious metals or oxidized metals can dramatically enhance the separated efficiency of

electron hole pairs in semiconductor materials, inhibit their recombination, and thus greatly improve their photocatalytic efficiency [5–10]. Layered dichalcogenide using $TiO_2$ as a photoactive material has great application value in terms of photocatalytic performance and supercapacitors [11,12]. Due to its electronic structure and narrow bandgap [13,14], $TiO_2$ can act as a photocatalyst, but its electron–hole pairs tend to recombine [15]. $TiO_2$ can only absorb UV radiation because of its bandgap energy of 3.2 eV. However, $MoS_2$ has a compatible energy band structure with $TiO_2$. By creating a heterostructure, $TiO_2$–$MoS_2$, it is possible to take advantage of $MoS_2$'s narrow bandgap characteristics to widen the light absorption radius and increase the light absorption intensity. Furthermore, by forming an internal electric field, the photogenerated current carrier recombination can be effectively inhibited [16–28].

Furthermore, noble metals such as platinum (Pt), silver (Ag), and gold (Au) can also be incorporated into titanium oxide, because precious metal nanoparticles can decrease the rapid recombination of photogenerated charge carriers, thus enabling the use of visible light [29–31]. By lowering photogenerated charge carriers, electrons of $TiO_2$ are transferred from the CB to the noble metal nanoparticles [32], increasing UV activity. Surface plasmon resonance effect and charge separation, which transmit photoexcited electrons from metal nanoparticles to $TiO_2$ CB, can explain photoactivity in the visible region of the electromagnetic spectrum [33]. Acquiring heterostructures by coupling two or more materials with distinct qualities makes it possible to enhance the photocatalytic activity of the system [34,35].

Based on the above analysis, it is possible that the components of this ternary structure may have synergetic advantages and improve visible light activation [36–39]. Herein, in this paper, a series of (M(M=Pt, Ag and Au)-$TiO_2$)@$MoS_2$ ternary heterojunction nanocomposites were prepared by the hydrothermal and deposition–precipitation methods, and the catalytic activity of the nanomaterials was investigated systematically under visible light irradiation with Rh B as the target contaminant. The experimental results showed that Pt (5 wt%) and $TiO_2$ (15 wt%) co-modified $MoS_2$ ternary heterojunction nanocomposites achieved the highest photocatalytic activity, with the maximum value of 99.0%, compared to 93% for $TiO_2$@$MoS_2$, 96.5% for (Ag-$TiO_2$)@$MoS_2$, and 97.8% for (Au-$TiO_2$) @$MoS_2$ within 9 min. The (Pt-$TiO_2$)@$MoS_2$ ternary heterojunction nanocomposites provided an effective path of photoexcited electrons from $TiO_2$ to surface-decorated Pt NPs using $MoS_2$ and internal Pt NPs as bridges, which greatly promoted electron transfer, reduced system overpotential, and led to more reactive areas being activated. The advanced structure of (Pt-$TiO_2$)@$MoS_2$ improved both light harvesting and electron transfer in photocatalytic composites, contributing to remarkable catalytic effectiveness and extended durability for the photodegradation of Rhodamine B (Rh B). In-depth discussions of the potential growth and photocatalytic mechanism, which will help improve the energy and environmental fields, are included.

## 2. Results and Discussion

*Characterization of the Synthesized Materials*

Using X-ray diffractometry, the crystallographic structures of $MoS_2$, $TiO_2$, (Pt-$TiO_2$)@$MoS_2$, (Ag-$TiO_2$)@$MoS_2$, and (Au-$TiO_2$)@$MoS_2$ samples were examined, and the characterization results are shown in Figure 1. $MoS_2$ has crystal planes (002), (100), (103), and (110), and in the diffractogram, it can be seen that (002), (100), (110), and (103) of $MoS_2$ have diffraction peaks at 13.03, 32.86, 37.78, and 58.04, respectively, which are consistent with the standard card (PDF 73-1508) [40]. Likewise, (200), (105), (211), (204), (101), and (004), and planes of $TiO_2$ were observed. The primary diffraction peaks of $TiO_2$ at 25.25, 35.99, 48.02, 53.94, 55.07, and 62.66 are indexed to the (200), (105), (211), (204), (101), and (004) planes of $TiO_2$, respectively [41]. They were in agreement with the $TiO_2$ anatase phase (PDF 99-0008). The presence of Pt, Ag, and Au, which are represented by a modest intensity peak at ca. 38° [42], corresponded to Pt(111), Ag(111), and Au (111), respectively. The XRD patterns of (Pt-$TiO_2$)@$MoS_2$, (Ag-$TiO_2$)@$MoS_2$, and (Au-$TiO_2$)@$MoS_2$ samples proved the

obvious peaks corresponding to the existence of $MoS_2$, $TiO_2$, and Pt, Ag, and Au, which showed that (Pt-$TiO_2$)@$MoS_2$, (Ag-$TiO_2$)@$MoS_2$, and (Au-$TiO_2$)@$MoS_2$ nanocomposites were successfully prepared.

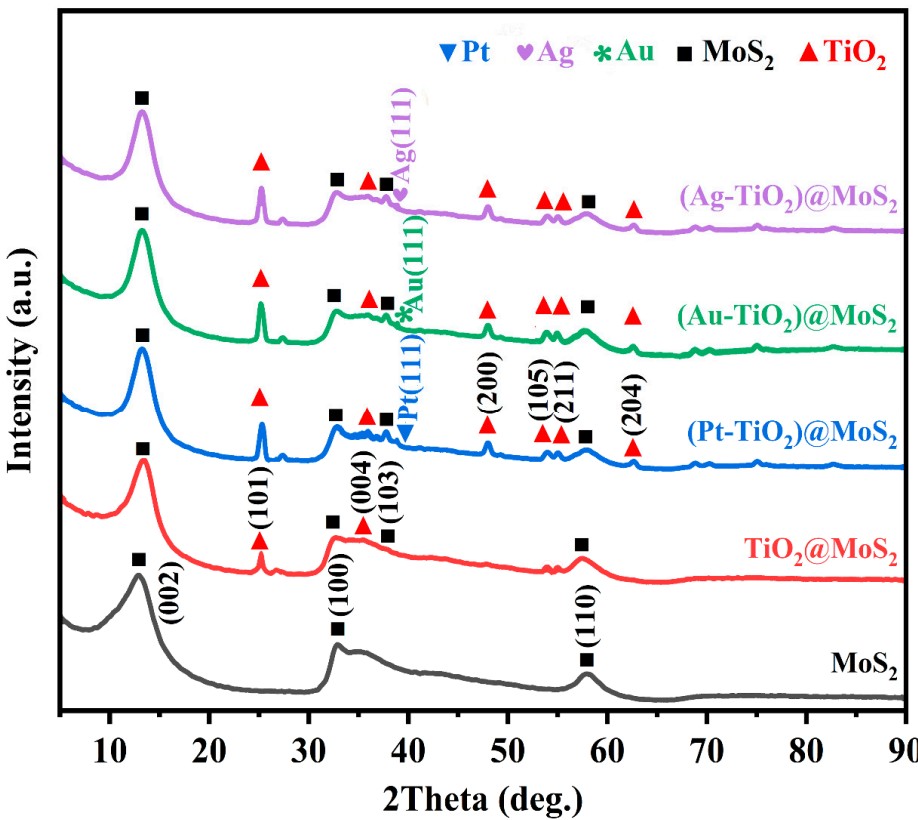

**Figure 1.** XRD patterns of $MoS_2$, $TiO_2$(15 wt%)@$MoS_2$, (Pt (5 wt%)-$TiO_2$(15 wt%))@$MoS_2$, (Ag (5 wt%)-$TiO_2$(15 wt%))@$MoS_2$, and (Au (5 wt%)-$TiO_2$(15 wt%))@$MoS_2$.

SEM and TEM were used to study the morphology of diverse samples.

Figure 2b,d shows that (Pt (5 wt%)-$TiO_2$(15 wt%))@$MoS_2$ is composed of dense sheets of $MoS_2$, P25 NPs, and Pt NPs, with disordered dispersion and no obvious change in morphology compared with pure $MoS_2$ (Figure 2a). Specifically, P25 NPs and Pt NPs are evenly distributed on the surface of $MoS_2$. A proportion of the P25 and Pt are completely encased in nanosheets, and the surface heterojunctions of $MoS_2$, P25, and Pt are achieved. This structure is conducive to photogenerated electron transfer among $MoS_2$, P25, and Pt, and it is convenient to separate the charge during the photocatalytic process.

As shown in Figure 2e, the HRTEM image of (Pt (5 wt%)-$TiO_2$(15 wt%))@$MoS_2$ indicates that the lattice fringes with d-spacing of 0.64, 0.35, and 0.22 nm correspond to the (002) lattice plane of $MoS_2$, (101) lattice plane of $TiO_2$ [43], and (111) lattice plane of Pt, respectively. Moreover, due to the beneficial contact between $MoS_2$, $TiO_2$, and Pt, photoinduced electrons on $MoS_2$ can be transferred quickly to Pt NPs via the $TiO_2$ nanocrystal bridge, so that the charge transfer distance is reduced and the electrons and holes are separated, which can increase the photocatalysis efficiency [36]. In addition, the element mapping images in Figure 2f–k prove the even dispersion and coexistence of Mo, S, Ti, O, and Pt.

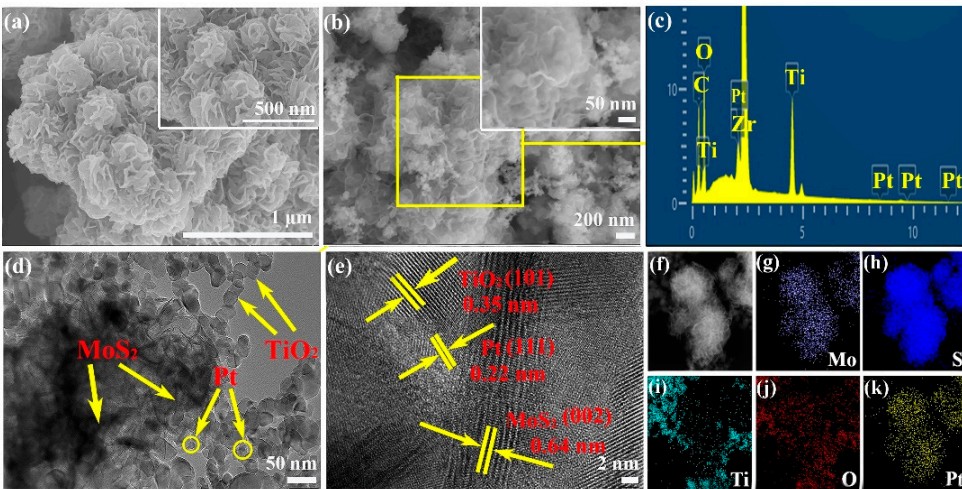

**Figure 2.** SEM images of (**a**) MoS$_2$ and (**b**) (Pt (5 wt%)-TiO$_2$(15 wt%))@MoS$_2$ and EDS patterns (**c**) corresponding to the SEM images of (**b**); TEM images of (**d**) (Pt (5 wt%)-TiO$_2$(15 wt%))@MoS$_2$. HRTEM image of (**e**) (Pt (5 wt%)-TiO$_2$(15 wt%))@MoS$_2$; (**f**–**k**) corresponding elemental mapping images of Mo, S, Ti, O, and Pt.

As shown in Figure 3e, HRTEM images of (Ag (5 wt%)-TiO$_2$(15 wt%))@MoS$_2$ indicate that the lattice fringes with d-spacing of 0.64, 0.35, and 0.23 nm correspond to the (002) lattice plane of MoS$_2$, (101) lattice plane of TiO$_2$ [43], and (111) lattice plane of Ag, respectively. Figure 3a shows the SEM image of (Ag (5 wt%)-TiO$_2$(15 wt%))@MoS$_2$, which shows a good dispersion. In addition, the element mapping images in Figure 3f–k prove the uniform distribution and coexistence of Mo, S, Ti, O, and Ag.

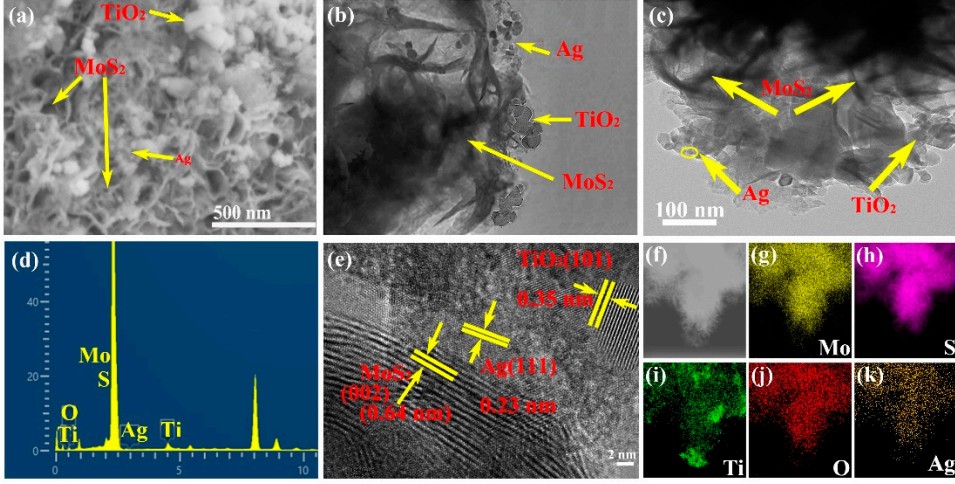

**Figure 3.** SEM images of (**a**) (Ag (5 wt%)-TiO$_2$(15 wt%))@MoS$_2$ and EDS pattern (**d**) corresponding to the SEM image of (**a**); TEM images of (Ag (5 wt%)-TiO$_2$(15 wt%))@MoS$_2$ (**b**,**c**) HRTEM image of (**e**) (Ag (5 wt%)-TiO$_2$(15 wt%))@MoS$_2$; (**f**–**k**) corresponding elemental mapping images of Mo, S, Ti, O, and Ag.

In Figure 4e, HRTEM images of (Au (5 wt%)-TiO$_2$(15 wt%))@MoS$_2$ indicate that the lattice fringes with d-spacing of 0.64, 0.35, and 0.235 nm correspond to the (002) lattice plane of MoS$_2$, (101) lattice plane of TiO$_2$ [43], and (111) lattice plane of Au, respectively. Figure 4a shows the SEM image of (Au (5 wt%)-TiO$_2$(15 wt%))@MoS$_2$, which shows a good dispersion. In addition, the element mapping images in Figure 4f–k prove the uniform distribution and coexistence of Mo, S, Ti, O, and Au.

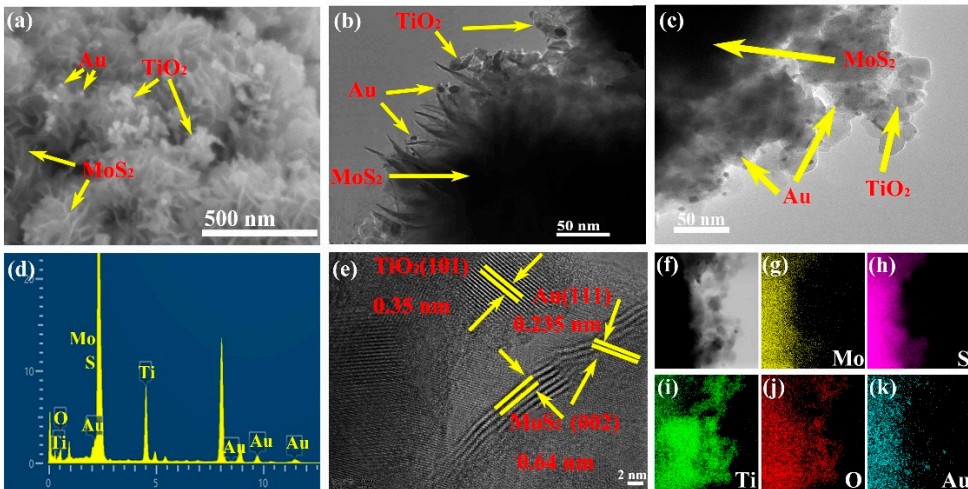

**Figure 4.** SEM images of (**a**) (Au (5 wt%)-TiO$_2$(15 wt%))@MoS$_2$ and EDS pattern (**d**) corresponding to the SEM image of (**a**); TEM images of (Au (5 wt%)-TiO$_2$(15 wt%))@MoS$_2$ (**b**,**c**); HRTEM image of (**e**) (Au (5 wt%)-TiO$_2$(15 wt%))@MoS$_2$; (**f**–**k**) corresponding elemental mapping images of Mo, S, Ti, O, and Au.

XPS measurement was used to analyze the valence and surface composition of the (Pt (5 wt%)-TiO$_2$ (15 wt%))@MoS$_2$ photocatalyst. The elements of O, Ti, Mo, S, and Pt were discovered in the survey spectra (see Figure 5a). TiO$_2$ peaks were seen at about 464.5 eV (Ti 2p1/2), 458.7 eV (Ti 2p3/2) (Figure 5a) and 529.88 eV, and 531.4 eV (O 1s) (see Figure 5b). As the flawed state, Ti$^{3+}$ can curb photogenerated electron–hole pair recombination and speed up separation of charge [36]. Figure 5c shows the peaks at binging energies of 231.5 eV and 228.4 eV that can be ascribed to Mo 3d3/2 and Mo 3d5/2 of Mo$^{4+}$, respectively. As can be seen from Figure 5d, S 2p1/2 and S 2p3/2 spectra of MoS$_2$ have two peaks at 162.8 eV and 161.58 eV [44,45]. The major peaks in Figure 5f, which correspond to core electrons of Pt 4f7/2 and Pt 4f5/2 with binding energies of 71.7 and 74.9 eV (difference VE 3.2 eV), respectively [46], indicate that the formed Pt is in a metallic state. Therefore, TiO$_2$ and Pt were successfully incorporated into MoS$_2$, as further demonstrated by the XPS spectra.

In order to understand the capture, migration, and separation of carriers in the samples, photoluminescence (PL) spectra of all samples were measured at room temperature at the simulation wavelength of 328 nm, and the outcomes are depicted in Figure 6. As seen in Figure 6, PL spectral line peaks of the five samples were similar, but with different intensities. In the PL line, the excitation peak at 400 nm can be ascribed to band emission from free exciton recombination. The photoluminescence spectra can reveal the recombination efficiency between photoelectrons and holes to a certain extent, because the secondary recombination between them will be accompanied by fluorescence emission. The stronger the fluorescence intensity that is produced, the more recombination between the electrons and holes that are produced and the shorter the carrier lifetime and vice versa. As can be seen from Figure 6, pure MoS$_2$ had a strong fluorescence emission, while the fluorescence intensities of TiO$_2$(15 wt%)@MoS$_2$, (Pt(5 wt%)-TiO$_2$(15 wt%))@MoS$_2$, (Ag(5 wt%)-TiO$_2$(15 wt%))@MoS$_2$, and (Au (5 wt%)-TiO$_2$(15 wt%)))@MoS$_2$ were significantly reduced. The results indicated that when TiO$_2$, Pt, Ag, and Au NPs are supported on the surface of MoS$_2$, the separation rate of the photoelectron–hole pairs of MoS$_2$ is significantly improved, which is obviously beneficial to improving the photocatalytic performance. However, compared with the luminescence intensity of TiO$_2$(15 wt%)@MoS$_2$, (Pt(5 wt%)-TiO$_2$(15 wt%))@MoS$_2$ had stronger luminescence intensity, which may have been due to the stronger excitonic PL signal and the higher surface oxygen vacancy and defect concentration. Furthermore, during photocatalytic activity, oxygen vacancies and defects can act as centers to capture photoinduced electrons, therefore inhibiting photoinduced electrons and holes from recombining. Additionally, oxygen vacancies can facilitate

oxygen adsorption, resulting in a strong interaction between photoinduced electrons bound by oxygen vacancies and adsorbed oxygen. This suggests that the binding of photoinduced electrons of oxygen vacancies can result in the capture of photoinduced electrons of adsorbed oxygen and oxygen radical groups at the same time. As a result, oxygen vacancies and defects are in favor of photocatalytic processes, because oxygen is active in promoting the oxidation of organic substances [47].

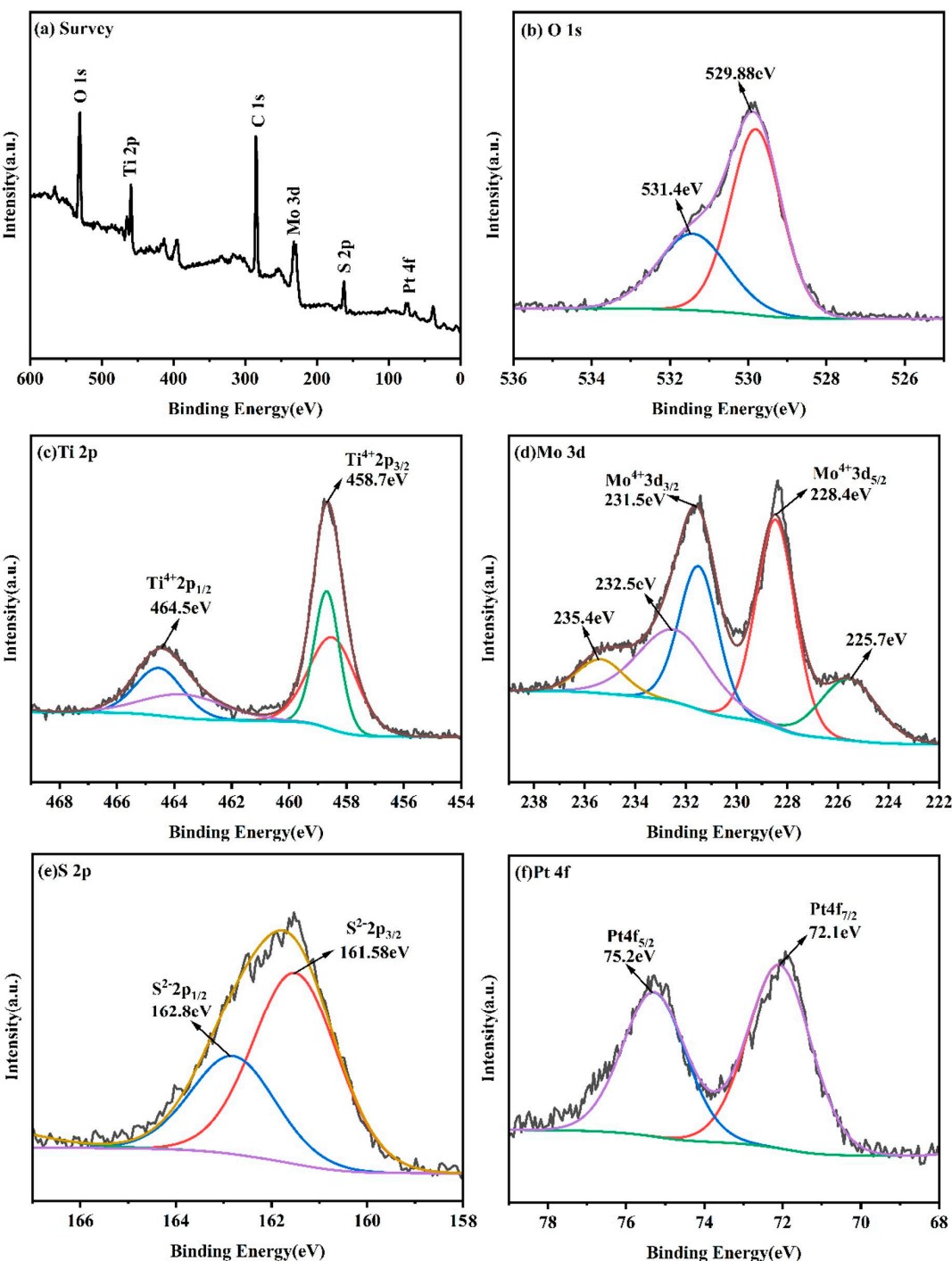

**Figure 5.** XPS spectra of (Pt (5 wt%)-TiO$_2$(15 wt%))@MoS$_2$: (**a**) survey spectrum, (**b**) O1s, (**c**) Ti2p, (**d**) Mo 3d, (**e**) S 2p, and (**f**) Pt 4f.

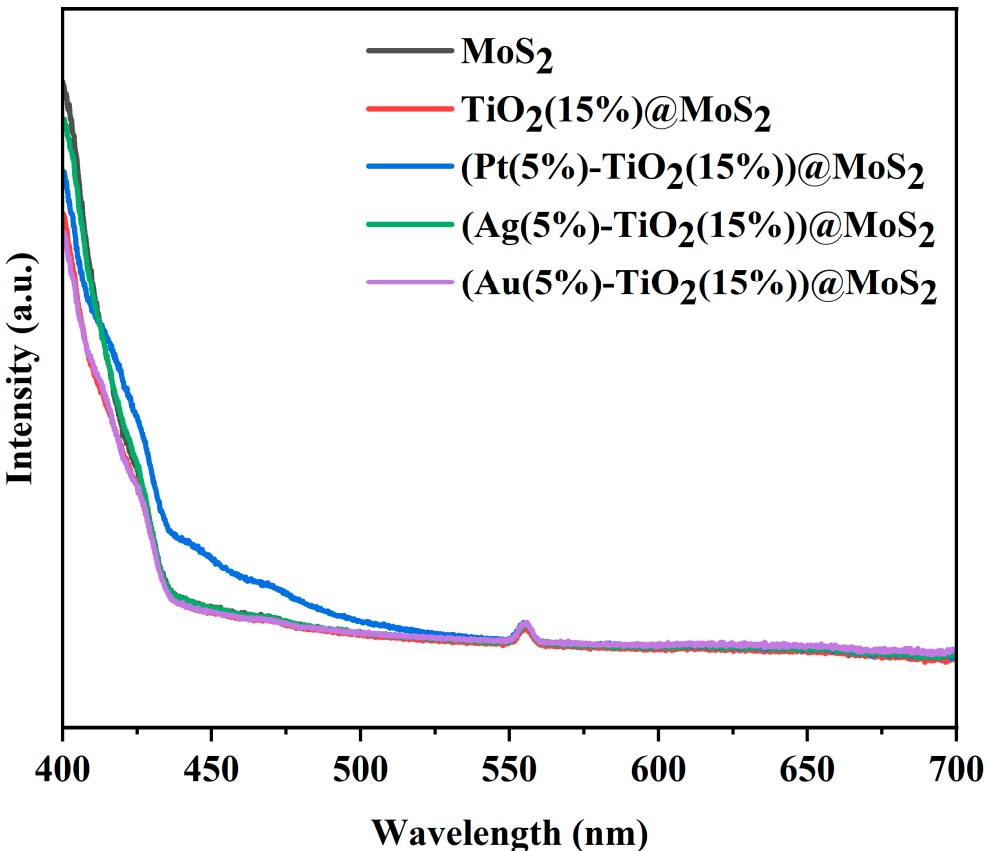

**Figure 6.** PL emission spectra of $MoS_2$, $TiO_2$(15 wt%)@$MoS_2$, (Pt (5 wt%)-$TiO_2$(15 wt%))@$MoS_2$, (Ag (5 wt%)-$TiO_2$(15 wt%))@$MoS_2$, and (Au (5 wt%)-$TiO_2$(15 wt%))@$MoS_2$.

The FT-IR of $MoS_2$, $TiO_2$(15 wt%))@$MoS_2$, (Pt(5 wt%)-$TiO_2$(15 wt%))@$MoS_2$, (Ag(5 wt%)-$TiO_2$(15 wt%))@$MoS_2$, and (Au(5 wt%)-$TiO_2$(15 wt%))@$MoS_2$ are shown in Figure 7. Compared to pure $MoS_2$, the chemical skeleton of the other samples was similar to $TiO_2$(15 wt%))@$MoS_2$. In addition, the peak strengths of $TiO_2$(15 wt%))@$MoS_2$, (Pt(5 wt%)-$TiO_2$(15 wt%))@$MoS_2$, (Ag(5 wt%)-$TiO_2$(15 wt%))@$MoS_2$, and (Au(5 wt%)-$TiO_2$(15 wt%))@$MoS_2$ nanocomposites were significantly enhanced compared with $MoS_2$, indicating that Ti-O was successfully doped into $MoS_2$, which was consistent with the results of XPS. In the infrared spectrum, the absorption peak was wide and gentle at the wavelength of 3000–3500 cm$^{-1}$, which corresponded to the stretching vibration peak of the $H_2O$ molecule adsorped on the surface of the nanoparticles. The absorption peak at 1700–1200 cm$^{-1}$ represented the stretching vibration absorption of C=S and C-H. The wide absorption band of 500–900 cm$^{-1}$ was caused by the bending vibration of Ti−O−Ti bonds in $TiO_2$ NPs [48]. The results of FT-IR showed that (M (M=Pt, Ag, Au)-$TiO_2$)@$MoS_2$ had no other chemical structure.

Figure 8a displays the UV–Vis DRS spectra of $MoS_2$, $TiO_2$(15 wt%))@$MoS_2$, (Pt(5 wt%)-$TiO_2$(15 wt%))@$MoS_2$, (Ag (5 wt%)-$TiO_2$(15 wt%)) @$MoS_2$, and (Au (5 wt%)-$TiO_2$(15 wt%)) @$MoS_2$. As can be seen from Figure 8, pure $MoS_2$ had strong absorption in both the ultraviolet and visible spectra. In comparison to pure $MoS_2$, when $TiO_2$, Pt, Ag, and Au nanoparticles were loaded, the absorption of $TiO_2$(15 wt%))@$MoS_2$, (Pt(5 wt%)-$TiO_2$(15 wt%))@$MoS_2$, (Ag(5 wt%)-$TiO_2$ (15 wt%))@$MoS_2$, and (Au (5 wt%)-$TiO_2$(15 wt%))@$MoS_2$ became stronger in the visible region and the absorption edge was obviously redshifted, which increased the utilization rate of visible light and indicated that more electron–hole pairs would be generated under the excitation of visible light. This would obviously be beneficial to photocatalysis. Figure 8b shows the $MoS_2$, $TiO_2$(15 wt%))@$MoS_2$, (Pt(5 wt%)-$TiO_2$(15 wt%))@$MoS_2$, (Ag(5 wt%)-$TiO_2$(15 wt%))@$MoS_2$, and (Au(5 wt%)-$TiO_2$(15 wt%))@$MoS_2$ band gap diagram. It can be seen in Figure 8 that the band gap widths of $MoS_2$, $TiO_2$(15 wt%))@$MoS_2$,

(Pt(5 wt%)-TiO$_2$(15 wt%))@MoS$_2$, (Ag(5 wt%)-TiO$_2$(15 wt%))@MoS$_2$, and (Au(5 wt%)-TiO$_2$(15 wt%))@MoS$_2$ were 1.26 eV, 1.16 eV, 0.55 eV, 1.05 eV, and 0.77 eV, respectively. Compared with MoS$_2$, TiO$_2$(15 wt%))@MoS$_2$, (Ag(5 wt%)-TiO$_2$(15 wt%))@MoS$_2$, and (Au(5 wt%)-TiO$_2$(15 wt%))@MoS$_2$, the (Pt(5 wt%)-TiO$_2$(15 wt%))@MoS$_2$ had the smallest band gap, which clearly demonstrated the activity of (Pt(5 wt%)-TiO$_2$(15 wt%))@MoS$_2$ under visible light irradiation. This corresponded to the catalytic results of five samples in the photocatalytic degradation of Rh B, which are described later.

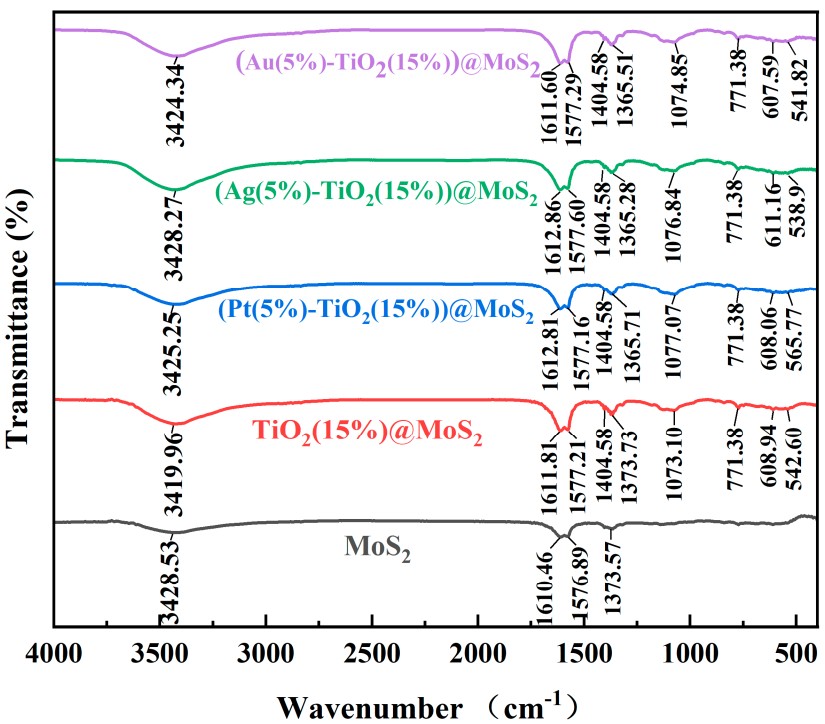

**Figure 7.** FT-IR spectra of MoS$_2$, TiO$_2$(15 wt%))@MoS$_2$, (Pt (5 wt%)-TiO$_2$(15 wt%))@MoS$_2$, (Ag (5 wt%)-TiO$_2$(15 wt%))@MoS$_2$, and (Au (5 wt%)-TiO$_2$(15 wt%))@MoS$_2$.

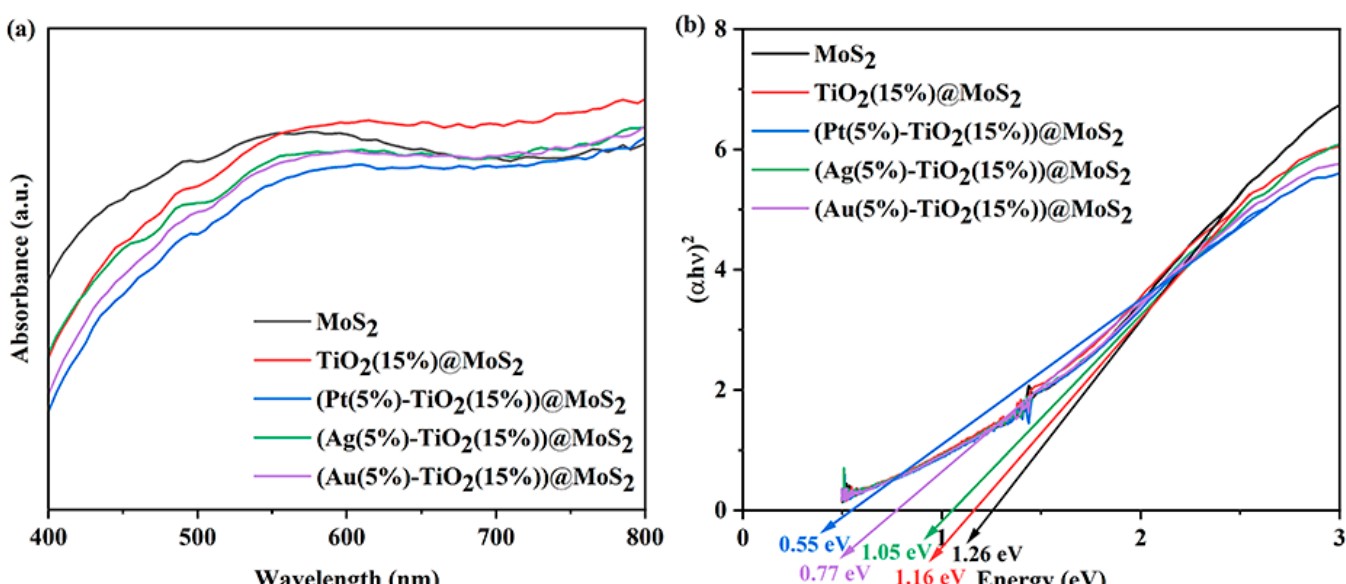

**Figure 8.** (**a**) UV–Vis DRS spectra and (**b**) the corresponding plot analysis of the optical band gap of MoS$_2$, TiO$_2$(15 wt%))@MoS$_2$, (Pt (5 wt%)-TiO$_2$(15 wt%))@MoS$_2$, (Ag (5 wt%)-TiO$_2$(15 wt%))@MoS$_2$, and (Au (5 wt%)-TiO$_2$(15 wt%))@MoS$_2$.

The result is shown in Figure 9. Under chopped light illumination, the corresponding responses (I-t) cycles were recorded at 0.6 V vs. Ag/AgCl. The photocurrent density also increased with the increase of Pt-content loading on the $TiO_2$@$MoS_2$ surface, demonstrating that the presence of more active points (Pt-$TiO_2$ heterojunctions) is conducive to reducing the recombination rate of photogenerated electron–hole pairs and promoting the transfer of photo-generated carriers. Nevertheless, the photocurrent densities of (Pt (2 wt%)-$TiO_2$(15 wt%))@$MoS_2$, (Au (5 wt%)-$TiO_2$(15 wt%))@$MoS_2$, and (Ag (5 wt%)-$TiO_2$(15 wt%))@$MoS_2$ were lower than that of (Pt (5 wt%)-$TiO_2$(15 wt%))@$MoS_2$. The optimum (Pt(5 wt%)-$TiO_2$(15 wt%))@$MoS_2$ nanocomposite had a higher photocurrent intensity than the pure $MoS_2$ sample. For all the electrodes tested, a rapid and even photocurrent response was observed for each opening–closure event, indicating that the test samples were well recycled. The photocurrent is largely dependent on the photo-separation efficiency of the electron–hole pairs at the electrode [49]. At the photocatalyst/electrolyte interface, photogenerated holes were transferred, and photogenerated electrons were simultaneously transferred to the back contact [50]. The higher photocurrent response indicated that (Pt (5 wt%)-$TiO_2$ (15 wt%)@$MoS_2$ was the best option for the separation of charge and transfer to the electrolyte.

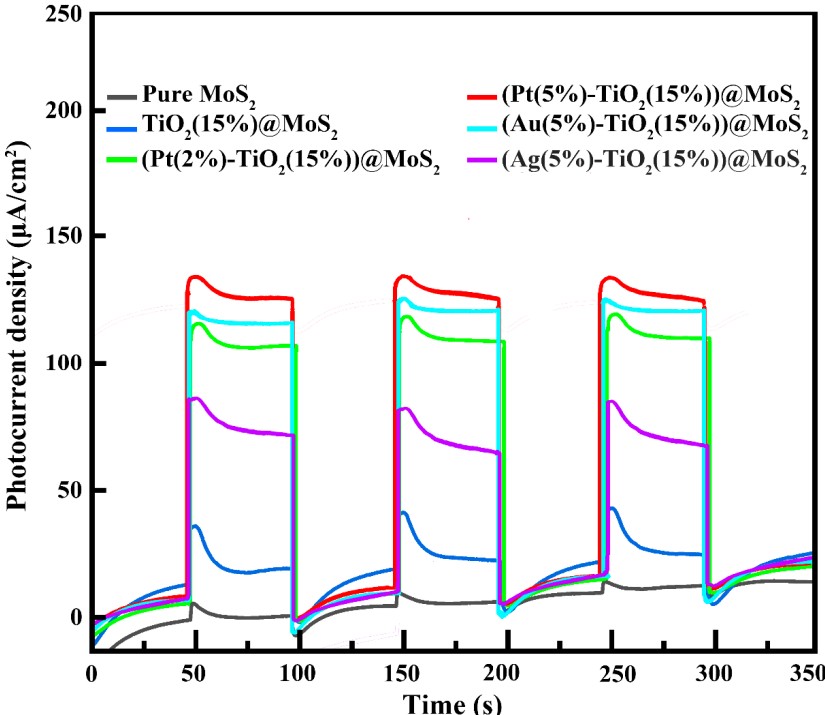

**Figure 9.** Chopped photocurrent response with light OFF/ON every 50 s of samples of pure $MoS_2$, $TiO_2$ (15 wt%)@$MoS_2$, (Pt (2 wt%)-$TiO_2$(15 wt%))@$MoS_2$, (Pt (5 wt%)-$TiO_2$(15 wt%))@$MoS_2$, (Au (5 wt%)-$TiO_2$(15 wt%))@$MoS_2$, and (Ag (5 wt%)-$TiO_2$(15 wt%))@$MoS_2$.

The nitrogen adsorption–desorption isotherms of (Pt(5 wt%)-$TiO_2$(15 wt%))@$MoS_2$, (Ag(5 wt%)-$TiO_2$(15 wt%))@$MoS_2$, (Au(5 wt%)-$TiO_2$(15 wt%))@$MoS_2$, $TiO_2$(15 wt%))@$MoS_2$, and $MoS_2$ showed stepwise adsorption behavior, as demonstrated in Figure 10. $N_2$ adsorption–desorption isothermal curves of the five samples were typical type IV adsorption–desorption curves. The specific surface areas of $MoS_2$, $TiO_2$@$MoS_2$ (Pt (5 wt%)-$TiO_2$(15 wt%))@$MoS_2$, (Ag(5 wt%)-$TiO_2$(15 wt%))@$MoS_2$, and (Au(5 wt%)-$TiO_2$(15 wt%))@$MoS_2$ were 12.62, 26.78, 32.85, 32.72, and 23.84 $m^2$/g, respectively. The specific surface area of (Pt(5 wt%)-$TiO_2$(15 wt%))@$MoS_2$ composite material was noticeably higher than those of (Ag(5 wt%)-$TiO_2$(15 wt%))@$MoS_2$, (Au(5 wt%)-$TiO_2$(15 wt%))@$MoS_2$, $TiO_2$(15 wt%))@$MoS_2$, and $MoS_2$, suggesting that the addition of Pt and $MoS_2$ contributed to the dispersion of $TiO_2$ nanoparticles and reduced agglomeration. These results matched the SEM image information, and

the specific surface area was improved. The photocatalysis of the nanocomposite was enhanced by increasing the active sites of the reaction.

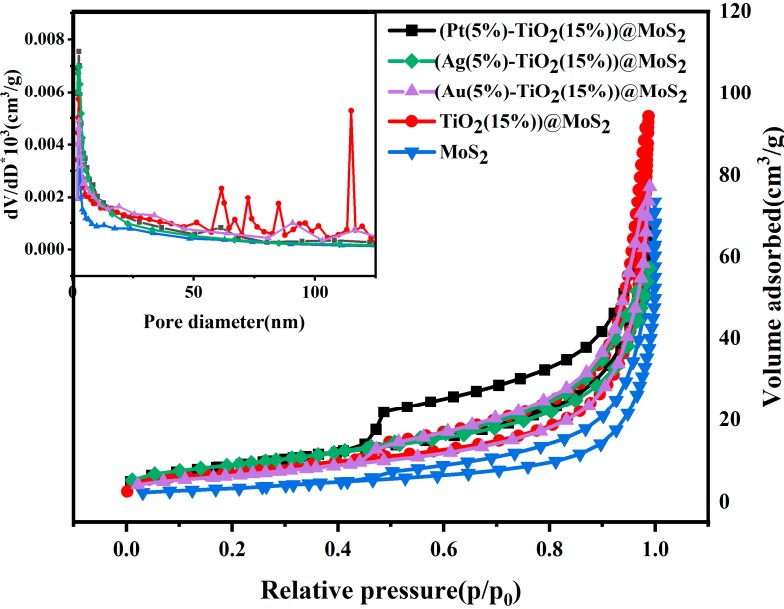

**Figure 10.** Nitrogen adsorption–desorption isotherms of (Pt(5 wt%)-TiO$_2$(15 wt%))@MoS$_2$, (Ag(5 wt%)-TiO$_2$(15 wt%))@MoS$_2$, (Au(5 wt%)-TiO$_2$(15 wt%))@MoS$_2$, TiO$_2$(15 wt%))@MoS$_2$, and MoS$_2$. The insert curve shows the pore size distribution.

According to the pore size analysis, the average pore size of (Pt(5 wt%)-TiO$_2$(15 wt%))@MoS$_2$ was 12.61 nm (see Table 1), which was smaller than (Au(5 wt%)-TiO$_2$(15 wt%))@MoS$_2$ (20.05 nm), TiO$_2$(15 wt%))@MoS$_2$ (21.78 nm), and MoS$_2$ (19.39 nm) and bigger than (Ag(5 wt%)-TiO$_2$(15 wt%))@MoS$_2$ (10.83 nm). The (Pt(5 wt%)-TiO$_2$(15 wt%))@MoS$_2$ nanocomposites were formed by the coupling of nano-TiO$_2$ and Pt NPs on the surface of MoS$_2$, which may have resulted in adhesion and aggregation of TiO$_2$ NPs and produced more mesopores or macropores, resulting in an increase in the average pore size of the (Pt(5 wt%)-TiO$_2$(15 wt%))@MoS$_2$ composites.

**Table 1.** BET specific surface areas and average pore sizes of (Pt(5 wt%)-TiO$_2$(15 wt%))@MoS$_2$, (Ag(5 wt%)-TiO$_2$(15 wt%))@MoS$_2$, (Au(5 wt%)-TiO$_2$(15 wt%))@MoS$_2$, TiO$_2$(15 wt%))@MoS$_2$, and MoS$_2$.

| Sample | (Pt(5 wt%)-TiO$_2$(15 wt%))@MoS$_2$ | (Ag(5 wt%)-TiO$_2$(15 wt%))@MoS$_2$ | (Au(5 wt%)-TiO$_2$(15 wt%))@MoS$_2$ | TiO$_2$(15 wt%))@MoS$_2$ | MoS$_2$ |
|---|---|---|---|---|---|
| S$_{BET}$ (m$^2 \times$ g$^{-1}$) | 32.85 | 32.72 | 23.85 | 26.78 | 12.62 |
| Average pore size (nm) | 12.61 | 10.83 | 20.05 | 21.78 | 19.39 |

The degradation of Rh B was photo-catalyzed by a series of MoS$_2$, TiO$_2$@MoS$_2$, and (M(M=Pt, Ag and Au)-TiO$_2$)@MoS$_2$ in the presence of simulated solar light. The photocatalytic degradation rate and kinetics of Pt (5 wt%)-TiO$_2$ (15 wt%)@MoS$_2$ photocatalyst were compared to those of Ag (5 wt%)-TiO$_2$ (15 wt%)@MoS$_2$, Au (5 wt%)-TiO$_2$ (15 wt%)@MoS$_2$, TiO$_2$@MoS$_2$, P25, and pure MoS$_2$ in Figure 11. Figure 11a shows that the TiO$_2$(15 wt%)@MoS$_2$ composite performed best of the eight samples, and Figure 11c shows that the Pt (5 wt%)-TiO$_2$(15 wt%)@MoS$_2$ composite performed best of the eight samples. The results of the UV visible absorption spectra test showed that Pt (5 wt%) and TiO$_2$ (15 wt%) co-modified MoS$_2$ ternary heterojunction nanocomposites exhibited the highest photocatalytic activity, with the maximum value of 99.0% compared to 93%

with $TiO_2@MoS_2$, 96.5% with $(Ag-TiO_2)@MoS_2$, and 97.8% with $(Au-TiO_2)@MoS_2$ within 9 min. Others exhibited varying degrees of $C/C_0$ fluctuation for Rh B degradation. The following conclusions can be drawn from this result: Rh B was absorbed onto the surface of $(Pt-TiO_2)@MoS_2$ nanocomposite materials, and with an increase in the loading amount of $Pt-TiO_2$, the degradation rate was higher, providing more active sites for the degradation of Rh B [51]. P25, with its mixture of anatase and rutile phases, has been commonly applied as a reference photocatalyst for evaluating photocatalysis activity. From Figure 11b,d, (Pt (5 wt%)-$TiO_2$(15 wt%))@$MoS_2$ possessed the highest kapp ($0.50470$ $min^{-1}$) for Rh B as compared to (Ag(5 wt%)-$TiO_2$(15 wt%))@$MoS_2$ ($0.32706$ $min^{-1}$), (Au(5 wt%)-$TiO_2$(15 wt%))@$MoS_2$ ($0.35668$ $min^{-1}$), $TiO_2$(15 wt%)@$MoS_2$ ($0.29247$ $min^{-1}$), P25 ($0.27312$ $min^{-1}$), and $MoS_2$ ($0.20109$ $min^{-1}$), which showed that the photocatalytic activity of (Pt (5 wt%)-$TiO_2$(15 wt%))@$MoS_2$ exceeded that of (Ag (5 wt%)-$TiO_2$(15 wt%))@$MoS_2$, (Au (5 wt%)-$TiO_2$(15 wt%))@$MoS_2$, $TiO_2$(15 wt%)@$MoS_2$, P25, and pure $MoS_2$ for the degradation of Rh B. Moreover, degradation of Rh B by pure $MoS_2$ was low, and the reaction rate constant was very low. This showed that the degradation of Rh B is mainly caused by the adsorption of $MoS_2$. Following Pt deposition, the above graph indicates that the particles absorbed more visible light [52], which is a typical optical characteristic of $Pt-TiO_2$ [53,54]. In conclusion, it is suggested that the ternary structure may have synergistic effects and reduce the rate of recombination and/or enhance the activation of visible light [36–39]. To study the optical absorption characteristics of photocatalysts, the UV–visible absorption spectra of $MoS_2$, $TiO_2@MoS_2$, $(Pt-TiO_2)@MoS_2$, $(Ag-TiO_2)@MoS_2$, and $(Au-TiO_2)@MoS_2$ with various mass ratios were examined (Figure 11b). Pure $MoS_2$ had absorption bands clearly showing the edges at 680 nm, whereas $MoS_2$ did not generate charges when activated by UV light [55]. Due to the visible light response of $MoS_2$, the absorption characteristics of $(Pt-TiO_2)@MoS_2$ nanocomposites redshifted progressively as the $MoS_2$ content increased [56]. According to the findings, the existence of $MoS_2$ in the visible light area effectively extended the $TiO_2$ nanoparticles' visible light absorption [57,58].

The photocatalysis mechanism of $(Pt-TiO_2)@$ $MoS_2$ (Figure 12) was considered. Generally, $MoS_2$ has a lower conduction band (CB) and Fermi level than P25 [37]. When $MoS_2$ contacts P25 in $(Pt-TiO_2)@MoS_2$ nanocomposites, the upward shifting of the Fermi level of $MoS_2$ and the downward shifting of the Fermi level of P25 will result in an equilibrium state [37]. Consequently, $MoS_2$ has a higher CB than P25. The following is a potential mechanism for the separation and transfer of charge in $(Pt-TiO_2)@MoS_2$: when the system is illuminated, electrons are rapidly simulated from the valence band (VB) of $MoS_2$ to the charge band (CB), leaving holes in the VB. Because P25 has a lower CB than $MoS_2$, it is possible to quickly transfer the excited electrons to P25. After being transported to the catalyst's surface, the electrons are captured by the oxygen molecules in the aqueous solution to form active free radicals, such as superoxide radical anions ($\cdot O^{2-}$) and hydroxyl radicals ($\cdot OH$), which eliminate organic contaminants. At the same time, the remaining holes in the VB can oxidize water to form $\cdot OH$, which reacts with organic species [59]. As illustrated in Figure 12, photogenerated electrons and holes separately accumulate in the CB of $MoS_2$ and the VB of $TiO_2$. The EBCB (BCB: bottom of the conduction band) of $MoS_2$ has a more negative redox potential than $O_2/\cdot O^{2-}$. Photogenerated electrons in the CB of $MoS_2$ can decrease the absorbed $O_2$ into $\cdot O^{2-}$. In addition, the ETVB of $TiO_2$ (TVB: the top of the valence band) has a more positive oxidation potential than $OH/\cdot OH$. This indicates that the formation of $\cdot OH$ in the VB of $TiO_2$ is easy. Nevertheless, visible light is the primary energy of the available photons when simulating solar radiation, which implies that only a small number of photons can be used efficiently. Hence, $TiO_2$ produces few $\cdot OH$, which is consistent with experiments. In brief, the photogenerated electrons on $TiO_2$ migrate quickly and collect on $MoS_2$. The hydroxyl groups on $MoS_2$ are oxidized by holes to form $\cdot OH$, the essential oxidant for dye removal. At the same time, $h^+$ can also straight oxidize Rh B to form small, nontoxic molecules of $H_2O$ and $CO_2$.

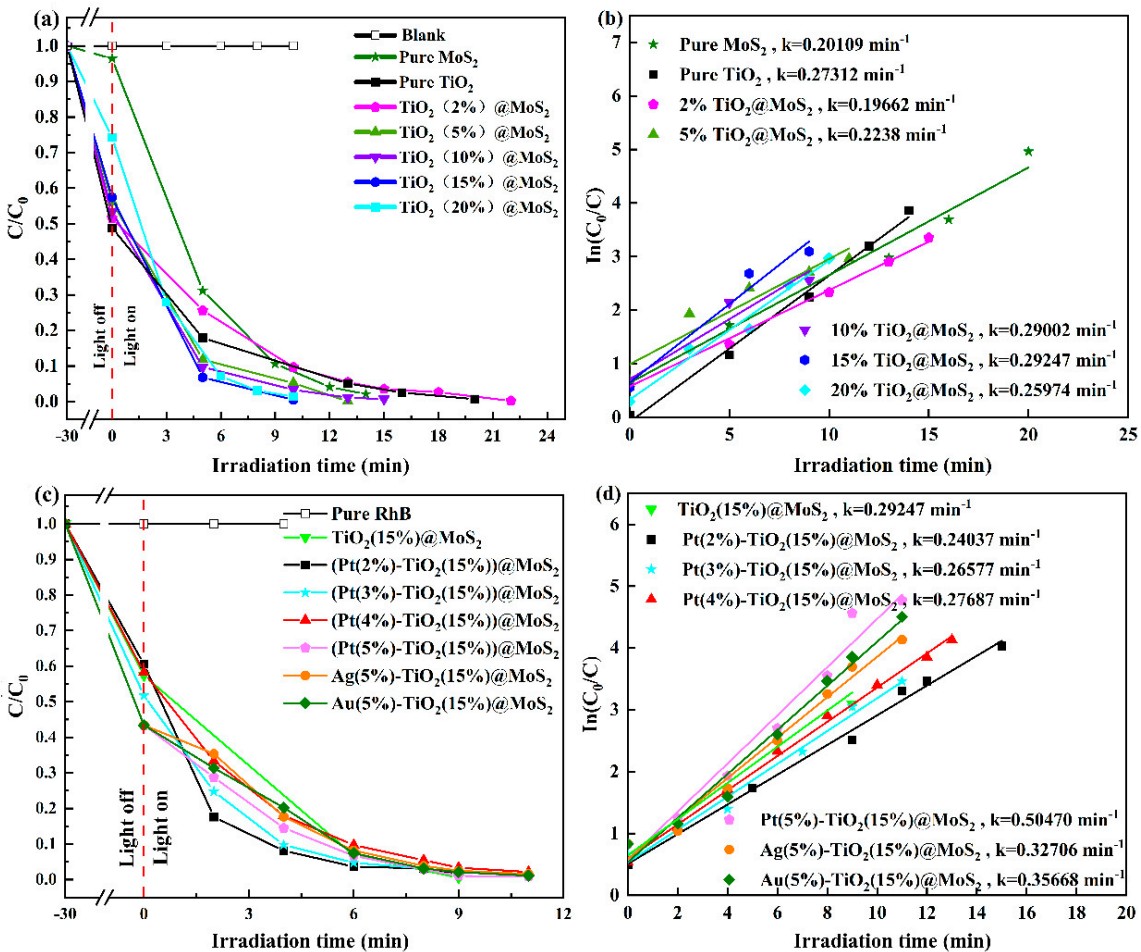

**Figure 11.** (**a**) Photocatalytic degradation of pure $MoS_2$ and $TiO_2$(2 wt%, 5 wt%, 10 wt%, 15 wt%, and 20 wt%) modified $MoS_2$ nanocomposites; (**b**) reaction kinetics corresponding to Figure 6a; (**c**) $TiO_2$(15 wt%)-$MoS_2$ and Pt(1 wt%, 2 wt%, 3 wt%, 4 wt%, and 5 wt%), Ag(5 wt%), Au(5 wt%) photocatalytic degradation of $TiO_2@MoS_2$ nanocomposites; (**d**) reaction kinetics corresponding to Figure 6c.

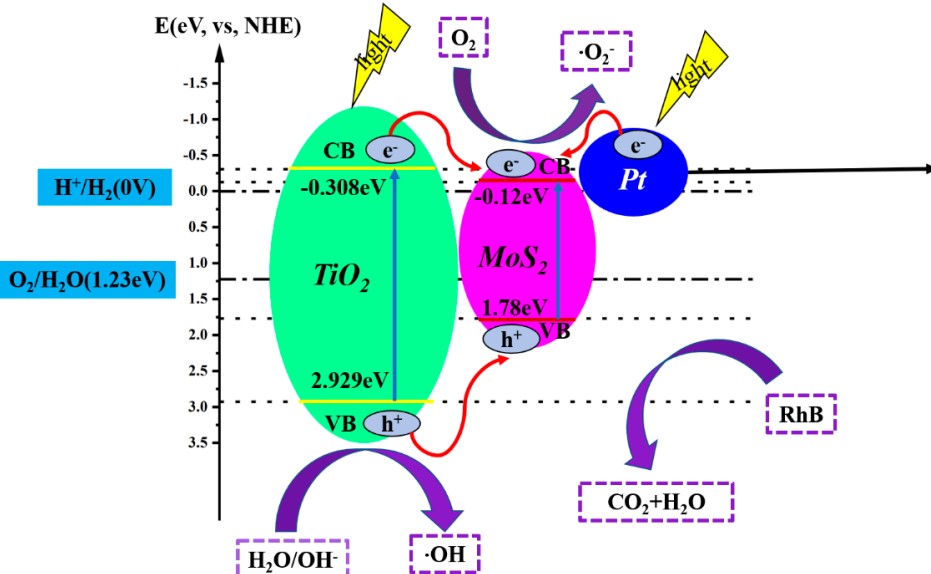

**Figure 12.** Schematic diagram of the energy band structure of (Pt-$TiO_2$)@$MoS_2$ nanocomposites and the proposed charge transfer mechanism.

## 3. Experiments

### 3.1. Materials

Titanium dioxide ($TiO_2$), hydrated sodium molybdate ($Na_2MoO_4 \cdot 2H_2O$, >99%), thiourea ($CH_4N_2S$, >99%), oxalic acid ($H_2C_2O_4$, >99%), sodium borohydride ($NaBH_4$, >99%), chloroplatinic acid ($H_2PtCl_6 \cdot 6H_2O$, >99%), silver nitrate ($AgNO_3$, >99%), $HAuCl_4 \cdot 3H_2O$ (>49%), L-ascorbic acid ($C_6H_8O_6$), and rhodamine B (Rh B) were acquired from Sinopharm Chemical Reagent Co., Ltd. (Shandong, China). All chemicals were of analytical grade and required no further purification for use. The water used in the experiment was deionized water.

### 3.2. Preparation of a Series of $MoS_2$, $TiO_2@MoS_2$, and $(Pt-TiO_2)@MoS_2$ Nanocomposites

#### 3.2.1. Preparation of $MoS_2$ Matrix Materials

$MoS_2$ was synthesized by the hydrothermal method [60–62]. Firstly, $CH_4N_2S$ (3.5 g), $Na_2MoO_4 \cdot 2H_2O$ (2.5 g), and $H_2C_2O_4$ (2.0 g) were successively dissolved in 350 mL DI water, followed by magnetic stirring for 0.5 h at room temperature. Next, the above solution was placed in an ultrasonic cleaner for 0.5 h. Moreover, the mixed solution was moved to a 500 mL Teflon stainless-steel autoclave and heated at 200 °C for 24 h. After that reaction and cooling at room temperature, the black product was centrifuged and washed six times alternatively with DI water and ethanol absolute through repeated re-dispersion and filtering. Finally, the material was dried at 85 °C for 12 h.

#### 3.2.2. Preparation of $TiO_2@MoS_2$ Nanocomposites

$TiO_2@MoS_2$ nanocomposites were obtained by the deposition–precipitation method. Using traditional synthesis, $MoS_2$ (0.170 g) and $TiO_2$ (0.030 g) were placed in two separate 100 mL beakers [21]. After adding 25 mL of ethanol absolute to each container, the mixture was sonicated for 30 min. Next, the $MoS_2$ solution and $TiO_2$ solution in the ultrasonic treated ethanol absolute were stirred by magnetic force, and the ultrasonic $MoS_2$ solution was slowly added to the dispersed $TiO_2$ solution at a speed of 5 mL·min$^{-1}$ with a syringe and stirred continuously until the dripping was finished. The grayish-black product was centrifuged after the reaction and washed six times with DI water and ethanol absolute through repeated re-dispersion and filtering. Finally, the material was dried for 12 h at 85 °C.

#### 3.2.3. Preparation of $(Pt-TiO_2)@MoS_2$, $(Ag-TiO_2)@MoS_2$, and $(Au-TiO_2)@MoS_2$ Nanocomposites

$(Pt-TiO_2)@MoS_2$ nanocomposites were obtained by the deposition–precipitation method [63]. $TiO_2@MoS_2$ nanocomposites (0.192 g) and $H_2PtCl_6 \cdot 6H_2O$ (8 mg) were poured into 50 mL ethanol absolute. The mixed solution was uniformly stirred for 1 h at room temperature. $C_6H_8O_6$ (15 mg) was mixed with the above solution with stirring for 2 h at 80 °C. Then, the acquired solution was centrifuged and washed six times alternatively with DI water and ethanol absolute through repeated redispersion and filtering. Finally, the material was dried for 12 h at 60 °C. The preparation of $(Au-TiO_2)@MoS_2$ nanocomposites was similar to that of $(Pt-TiO_2)@MoS_2$.

The $(Ag-TiO_2)@MoS_2$ nanocomposites were obtained by deposition–precipitation of Ag on the surface of $TiO_2@MoS_2$, on the basis of the method described by Naldoni group [64,65]. Using traditional synthesis, 500 mg of $TiO_2@MoS_2$ was scattered in water and stirred for 30 min. Then, the required amount of $AgNO_3$ was mixed with the above solution and stirred for 10 min. Next, $NaBH_4$ (aq) (10 mg in 10 mL of water) was added dropwise to the solution and left to react for 30 min. Finally, the acquired solution was centrifuged and washed six times alternatively with DI water and ethanol absolute through repeated redispersion and filtering and dried over night at 60 °C.

### 3.3. Characterization

The X-ray diffractometer (D8 Advance, Bruker, Billerica, MA, USA) was equipped with a Cu tube for producing Cu radiation (k = 1.5418 Å) and used to examine the crystal

forms of the composite photocatalyst. A Phi X-tool instrument was used to carry out the X-ray photoelectron spectroscopy (XPS) measurements. A scanning electron microscope (SEM) was used to examine the prepared photocatalyst morphology, internal structure, and composition. The transmission electron microscope (TEM, JEOL, JEM-2100F, Japan) and GeminiSEM300 were both from Germany. Using a TU-1810PC UV Vis spectrophotometer, diffuse reflectance spectra in the UV–Vis range were recorded. A surface area analyzer (BK112T) was used to characterize a specific area of the obtained samples. A 3H-2000PS1 isothermal nitrogen sorption analyzer from Beishide, China, produced the samples' nitrogen adsorption isotherms.

*3.4. Photocatalytic Measurement*

To investigate the photocatalysis properties of the obtained samples, 20 mg/L Rh B solution was used as the pollutant. The 20 mg nanocomposites tested were immersed in 100 mL of Rh B (20 mg/L$^{-1}$) aqueous solution to form a suspension. Magnetic stirring of the suspension was performed for 30 min in the dark to establish an adsorption–desorption equilibrium. A 300 W Xe lamp with a 420 nm UV-cut filter was then used as the light source. During irradiation, samples were taken at specified intervals and withdrawn from the mixture with a 0.22 μm syringe filter. The strength of the solution was measured at 554 nm by a TU-1810PC UV-visible spectrophotometer. Finally, the photocatalysis efficiency was calculated on the basis of $C/C_0 \times 100\%$, in which C and $C_0$ were the final and initial concentrations of the dyes.

## 4. Conclusions

Using XRD, SEM, TEM, XPS, PL, FTIR, UV–Vis DRS, and BET analyzer, the results confirmed that $TiO_2$ and M(Pt, Ag and Au) (NPs) were evenly distributed on $MoS_2$ nanosheets (NSs) to form (M(M=Pt, Ag and Au)-$TiO_2$)@$MoS_2$ nanocomposite heterojunctions. The photocatalytic degradation efficiency of Rh B was contrasted between a series of $MoS_2$, $TiO_2$@$MoS_2$, and (M(M=Pt, Ag and Au)-$TiO_2$)@$MoS_2$ nanocomposites with different compositions using a UV–Vis absorption spectrometer. The results showed that the (Pt(5 wt%)-$TiO_2$(15 wt%))@$MoS_2$ ternary heterojunction nanocomposites exhibited the highest photocatalysis activity, with the maximum value of 99.0%, compared to 93% for $TiO_2$(15 wt%)@$MoS_2$, 96.5% for (Ag(5 wt%)-$TiO_2$(15 wt%))@$MoS_2$, and 97.8% for (Au(5 wt%)-$TiO_2$(15 wt%))@$MoS_2$ within 9 min. The experiments showed that (Pt(5 wt%)-$TiO_2$(15 wt%))@$MoS_2$ surpassed $TiO_2$(15 wt%)@$MoS_2$, (Ag(5 wt%)-$TiO_2$(15 wt%))@$MoS_2$, and (Au(5 wt%)-$TiO_2$(15 wt%))@$MoS_2$ in photocatalytic degradation, and the (Pt(5 wt%)-$TiO_2$(15 wt%))@$MoS_2$ ternary heterojunction nanocomposites exhibited the highest photocatalytic activity of all of the samples. The advanced structure of (Pt-$TiO_2$)@$MoS_2$ improved both light harvesting and electron transfer in the photocatalytic composites, contributing to the remarkable catalytic effectiveness and extended durability in the photodegradation of Rh B. In-depth discussions of the potential growth and photocatalytic mechanism, which will help improve the energy and environmental fields, are included.

**Author Contributions:** Conceptualization, L.J., D.H., X.Y. (Xiude Yang) and X.Y. (Xiang Yang); Methodology, L.J., X.J., L.N., W.Z., Y.W. and X.Y. (Xiang Yang); Software, L.J., D.H., L.N., W.Z. and Y.W.; Validation, L.J., L.N., Y.W. and X.Y. (Xiang Yang); Formal analysis, L.J.; Investigation, L.J., L.N., W.Z., Y.W. and X.Y. (Xiang Yang); Resources, D.H., X.J., H.L., X.Y. (Xiude Yang), Q.L. and Z.G.; Data curation, L.J., Y.W. and X.Y. (Xiang Yang); Writing—original draft, L.J.; Writing—review & editing, L.J., D.H., X.J., H.L. and X.Y. (Xiude Yang); Supervision, D.H., X.J., H.L., Q.L. and Z.G.; Project administration, D.H., X.J., H.L., X.Y. (Xiude Yang), Q.L. and Z.G.; Funding acquisition, D.H. All authors have read and agreed to the published version of the manuscript.

**Funding:** This work was supported by the Scientific Research Foundation of Guizhou Province (ZSKH [2020] 1Y048, Key Field Project of Guizhou Province Education Ministry ([2020] 048) and Doctor Foun-dation of Zunyi Normal College (ZSBS [2018] 10), Innovation and Entrepreneurship Training Pro-gram for College Students in Guizhou Province (S202210664006). This work was also supported by the Key Laboratory of Zunyi City (SSKH [2015] 55).

**Acknowledgments:** This work was supported by the Scientific Research Foundation of Guizhou Province (ZSKH [2020] 1Y048, Key Field Project of Guizhou Province Education Ministry ([2020] 048) and Doctor Foundation of Zunyi Normal College (ZSBS [2018] 10), Innovation and Entrepreneurship Training Program for College Students in Guizhou Province (S202210664006). This work was also supported by the Key Laboratory of Zunyi City (SSKH [2015] 55).

**Conflicts of Interest:** The authors declare no conflict of interest.

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
