# Peer review of "Preparation and Study of Photocatalytic Properties of (M(M=Pt, Ag and Au)-TiO2)@MoS2 Nanocomposites"

_inorganics, doi:10.3390/inorganics11060258_

Round 1

Reviewer 1 Report

The authors performed an experimental investigation on the preparation and photocatalytic properties of 2 (M(M=Pt, Ag, and Au)-TiO2) @MoS2 nanocomposites. The authors synthesized a series of MoS2, TiO2@MoS2, (M(M=Pt, Ag, and Au)-TiO2)@MoS2 nano-9 composites with various compositions and investigated their photocatalytic performance.

In my opinion, this paper reports valuable results and deserves publication. However, some changes are necessary before doing so.

-In figures 6(a), 6(b), and 6(d), the title of the horizontal axis should be Irradiation and not “Irridiation.”

-In line 253, the authors write: “Figure 6c shows that the Pt (5 wt%)-TiO2(15 wt%) @MoS2 composite performs best of the eight samples.” It seems the authors made a typing mistake, and it should be Pt (2 wt%)-TiO2(15 wt%) @MoS2. If it is not a mistake, then they must explain their choice.

Author Response

1.For “In figures 6(a), 6(b), and 6(d), the title of the horizontal axis should be Irradiation and not “Irridiation.””.

I have made the modification, please review.

2.For "-In line 253, the authors write: “Figure 6c shows that the Pt (5 wt%)-TiO2(15 wt%) @MoS2 composite performs best of the eight samples.” It seems the authors made a typing mistake, and it should be Pt (2 wt%)-TiO2(15 wt%) @MoS2. If it is not a mistake, then they must explain their choice".

In line 253, the reason I came to this conclusion is that as shown in Figure 6c, in the early stage, Pt (2 wt%)-TiO2(15 wt%) @MoS2 degraded Rh B a little faster than Pt (5 wt%)-TiO2(15 wt%) @MoS2. However, in the later stage, the degradation rate of Rh B by Pt (5wt %)-TiO2(15wt %) @MoS2 was higher than that by Pt (2wt %)-TiO2(15wt %) @MoS2. Finally, in general, The degradation of Rh B by Pt (5 wt%)-TiO2(15 wt%) @MoS2 was better than that by Pt (2 wt%)-TiO2(15 wt%) @MoS2 within 9 min. And from Figure 6d, (Pt (5 wt%)-TiO2(15 wt%)) @MoS2 possesses the highest kapp (0.50470 min-1) for Rh B as-compared to that of (Ag (5 wt%)-TiO2(15 wt%)) @MoS2(0.32706 min-1), (Au (5 wt%)-TiO2(15 wt%)) @MoS2(0.35668 min-1), TiO2(15 wt%) @MoS2 (0.29247 min-1), P25 (0.27312 min-1) and pure MoS2 (0.20109 min-1), which shows that the photocatalytic activity of (Pt (5 wt%)-TiO2(15 wt%)) @MoS2 surpass the (Ag (5 wt%)-TiO2(15 wt%)) @MoS2, (Au (5 wt%)-TiO2(15 wt%)) @MoS2, TiO2(15 wt%)@ MoS2, P25 and pure MoS2 for the degradation of Rh B.

3.The above is my explanation and modification, I hope your criticism and guidance, thank you.

Reviewer 2 Report

1.      You need to represent a clear novelty of your work in the abstract portion, in the last paragraph of introduction and in the conclusion section.

2.      Compare your work with previous published one and show the difference. It will be better if you provide all these result in form of table.

3.      The introduction portion is not enough and informative. Please focus over this.

4.      The as-prepared materials were used for photodegradation which happen due to photogenerated holes. So, it will be better to provide the following characterization for justification this process.  

(a)    FTIR has not been given. Provide it.

(b)   For photocatalytic it is important to analyze the PL and DRS of materials.

(c)    Provide the elemental analysis to show the elemental composition of all materials.

(d)   Show the surface area and pore size of all materials before and after modification in form of table.

5.      Are you done one process in the valence band and you shown the effect of platinum in the conduction band and Pt have the tendency to generate more and more active site for the electron formation which successively perform the reduction process? Did you perform any reduction process in the CB or only oxidation process in the VB. If not then explain.

6.      Is the process containing any biproducts??

7.      Special focus toward scientific English, check it and correct the grammar, sentence fluency, verb etc.

8.      References need to be adopted over the journal style. Good luck

Author Response

1.I have made changes to the abstract, the last paragraph of the introduction and the conclusion for your review. Thank you for your advice.

2.I have made the modification for your review. Thank you for your advice.

3.I have made the modification for your review. Thank you for your advice.

4.FTIR, PL, DRS, BET data has been added for your review. Thank you for your advice.

5.Materials are excited by light to produce electrons and holes, and the recombination reaction between carriers occurs, and energy can be released in the form of heat and light energy, and oxidation reaction is triggered by valence band holes. Reduction reaction triggered by conduction electrons; Further catalytic reactions occur. Photocatalytic reactions should be a fusion of photochemical and thermal catalytic reactions, i.e. reactions that can only occur when light and catalyst work together. The holes can react with hydroxide ions and water adsorbed on the surface of catalyst particles to form ·OH. ·OH is a more active substance that can oxidize and mineralize many organic compounds indiscriminately, and is generally regarded as the main active oxide in photocatalysis. Thank you for your advice.

6.This enhanced photocatalytic activity of nanocomposite is due to its high light absorption and lower electron-hole recombination for the degradation of dyes. No harmful by-products are formed. Thank you for your advice.

7.I have made the modification for your review. Thank you for your advice.

8.I have made the modification for your review. Thank you for your advice.